# GAUSSIAN SPLATTING LUCAS-KANADE

**Liuyue Xie[1], Joel Julin[1], Koichiro Niinuma[2], László A. Jeni[1]**
[1]Carnegie Mellon University, [2]Fujitsu Research of America
{liuyuex,jjulin}@andrew.cmu.edu, laszlojeni@cmu.edu, kniinuma@fujitsu.com

## ABSTRACT

Gaussian Splatting and its dynamic extensions are effective for reconstructing 3D scenes from 2D images when there is significant camera movement to facilitate motion parallax and when scene objects remain relatively static. However, in many real-world scenarios, these conditions are not met. As a consequence, data-driven semantic and geometric priors have been favored as regularizers, despite their bias toward training data and their neglect of broader movement dynamics.

Departing from this practice, we propose a novel analytical approach that adapts the classical Lucas-Kanade method to dynamic Gaussian splatting. By leveraging the intrinsic properties of the forward warp field network, we derive an analytical velocity field that, through time integration, facilitates accurate scene flow computation. This enables the precise enforcement of motion constraints on warp fields, thus constraining both 2D motion and 3D positions of the Gaussians. Our method excels in reconstructing highly dynamic scenes with minimal camera movement, as demonstrated through experiments on both synthetic and real-world scenes. [1]

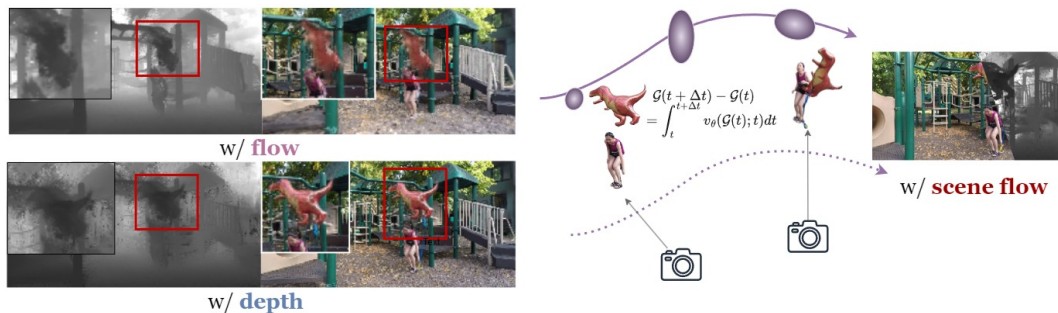

Figure 1: Data-driven depth and optical flow supervisions produce inaccurate geometries. Instead, we derive the analytical warp field of Gaussians to refine geometries and motions.

## 1 INTRODUCTION

The task of reconstructing dynamic 3D scenes from monocular video presents a significant challenge, particularly when constrained by limited camera movement. This limitation results in a lack of parallax and epipolar constraints, which are crucial for accurately estimating scene structures Gao et al. (2022); Schönberger & Frahm (2016). Implicit neural representation methods like NeRF have shown promise in mitigating this issue by modeling dynamic scenes using an implicit radiance field Mildenhall et al. (2021a); Chen et al. (2023); Li et al. (2021d). Explicit approaches like Gaussian splatting Kerbl et al. (2023), while more efficient and often improve rendering quality, face additional hurdles. Gaussian splatting relies on an explicit representation of the scene using volumetric Gaussian functions. This necessitates a high degree of accuracy in geometric understanding to ensure proper scene depiction. Consequently, despite advancements in dynamic Gaussian splatting techniques, the requirement for diverse viewing angles to effectively constrain the placement of these Gaussians remains a limiting factor. This inherent limitation restricts their effectiveness for capturing scenes from static viewpoints or dealing with objects exhibiting rapid and complex movements.

---

[1]Project page: https://gs-lk.github.io

Recent Gaussian splatting frameworks for modeling dynamic scenes commonly utilize a canonical Gaussian space as their foundation. This space is typically initialized using Structure from Motion (SfM) Micheletti et al. (2015) point clouds, with which each Gaussian is assigned attributes that describe its position, orientation, and lighting-dependent color. This canonical space acts as a reference point from which deformations are applied to represent the dynamic scene at different time steps. The process of deforming the canonical space to depict the scene at a particular time step is achieved through a warp field, often referred to as a forward warp field due to its function of warping the canonical representation forward to a specific point in time. This warp field is typically modeled using learnable Multi-Layer Perceptrons (MLPs) or offset values, as seen in frameworks like DeformableGS Yang et al. (2023) and Dynamic-GS Luiten et al. (2024). These approaches have laid the groundwork for dynamic Gaussian scene modeling.

Structural cues such as optical flow and monocular depth have been explored to regulate the temporal transitions of Gaussians. However, imposing structural constraints on a generic forward warp field, as seen in DeformableGS Yang et al. (2023), is challenging. Although many studies leverage depth and flow priors, they often fail to regularize motion through the warp field, resulting in sub-optimal trajectories and insufficient motion regularization. Additionally, employing time integration on an auxiliary network to predict the velocity field can introduce computational overhead and lack theoretical constraints. In contrast to this data-driven approach, we demonstrate that it is possible to derive velocity fields directly from a generic warp field by adapting the Lucas-Kanade method with a small motion constraint for Gaussian Splatting. This allows for effective time integration for structural comparisons. By regulating Gaussian motions through analytical velocity fields, we address the limitations of data-driven structural regularizations, enabling continuous learning of the warp field in a time domain without the need for additional learnable parameters.

In real-world captures, our approach outperforms state-of-the-art methods, including other dynamic Gaussian techniques. We demonstrate that our warp field regularization accommodates deformations in complex scenes with minimal camera movement, achieving results that are competitive with NeRF frameworks and enabling Gaussian splatting to model highly dynamic scenes from static cameras. To summarize, our approach offers two key advantages:

**Reduced Bias:** Methods relying solely on data-driven learning of warp fields can be biased towards the visible time steps, neglecting the overall movement of Gaussians. Our method, applicable to forward warp field techniques, ensures accurate Gaussian trajectories throughout the sequence.

**Improved Tractability:** The derived analytical solution for warp field velocities provides greater tractability compared to directly supervising the flow field as the difference in estimated deformations between consecutive frames.

## 2 RELATED WORKS

**Neural Radiance Fields (NeRFs).** NeRFs Mildenhall et al. (2021b) have demonstrated exceptional capabilities in synthesizing novel views of static scenes. Their approach uses a fully connected deep network to represent a scene's radiance and density. Since its introduction, NeRF has been extended to dynamic scenarios by several works Gafni et al. (2021); Park et al. (2021a;b); Pumarola et al. (2021); Tretschk et al. (2021b); Yu et al. (2023). Some of these methods leverage rigid body motion fields Park et al. (2021a;b) while others incorporate translational deformation fields with temporal positional encoding Pumarola et al. (2021); Tretschk et al. (2021b). Despite the advantages of NeRF and its dynamic extensions, they often impose high computational demands for both training and rendering.

**Gaussian Splatting.** The computational complexity of Neural Radiance Fields (NeRF) has driven the development of alternative 3D scene representation methods. 3D Gaussian Splatting (3DGS) Kerbl et al. (2023) has emerged as a promising solution, leveraging 3D Gaussians to model scenes and offering significant advantages over NeRF in terms of rendering speed and training efficiency.

While initially focused on static scenes, recent works Yang et al. (2023); Wu et al. (2024) have extended 3DGS to the dynamic domain. These approaches introduce a forward warp field that maps canonical Gaussians to their corresponding spacetime locations, enabling the representation of dynamic scene content. DeformableGS Yang et al. (2023) employs an MLP to learn positional, rotational, and scaling offsets for each Gaussian, creating an over-parameterized warp field that

captures complex spacetime relationships. 4DGaussians Wu et al. (2024) further refine this approach by leveraging hexplane encoding Fang et al. (2022); Cao & Johnson (2023) to connect adjacent Gaussians. DynamicGS Luiten et al. (2024) incrementally deforms Gaussians along the tracked time frame. Despite these advancements, existing dynamic 3DGS methods struggle with highly dynamic scenes and near-static camera viewpoints. To overcome these limitations, we propose a novel approach that grounds the warp field directly on approximated scene flow.

**Flow supervision.** Optical flow supervision has been widely adopted for novel view synthesis and 3D reconstruction. Dynamic NeRFs Li et al. (2021d; 2023; 2021a); Tretschk et al. (2021a); Wang et al. (2023); Chen et al. (2023) have explored implicit scene flow representation with an Invertible Neural Network, semi-explicit representation from time integrating a learned velocity field, and explicit analytical derivation from the deformable warp field. Flow-sup NeRF Wang et al. (2023) pioneered the use of analytical flow supervision in NeRF, laying the foundation for more precise and physically grounded modeling of dynamic scenes.

Several Gaussian Splatting works have begun exploring flow supervision. Motion-aware GS Guo et al. (2024) applies flow supervision on the cross-dimensionally matched Gaussians from adjacent frames, without explicitly accounting for the flow contributions of Gaussians to each queried pixel. Gao et al.Gao et al. (2024) on avatar rendering apply flow supervision for adjacent frames after rendering the optical flow map with $\alpha$-blending Wallace (1981). However, while these approaches introduce data-driven flow supervision, they do not effectively regularize the motion through the warp field, leading to suboptimal trajectories and insufficient motion regularization. This limitation makes them suitable only for enforcing short-term geometric consistency, while their performance deteriorates for out-of-distribution time steps Aoki et al. (2019); Li et al. (2021b). In the context of dynamic Gaussian Splatting, we propose to regularize the deformable warp field through analytical time integration to ensure consistent geometric relationships across time steps.

## 3 PRELIMINARY

**Dynamic Gaussian Splatting.** Dynamic 3D Gaussian Splatting frameworks follow a similar optimization pipeline as static Gaussian Splatting. Each Gaussian is defined by parameters for its mean $\boldsymbol{\mu}$, covariance $\Sigma$, and spherical harmonics $SH$ color coefficients. To project Gaussians from 3D to 2D, we calculate the view space covariance matrix as follows:

$$\Sigma' = JV\Sigma V^T J^T, \tag{1}$$

where $J$ is the Jacobian of the projective transformation and $V$ is the viewing transform. To represent a scene's radiance, the covariance $\Sigma$ is decomposed into scaling matrices $S$ and rotation matrices $R$ as: $\Sigma = RSS^T R^T$. This decomposition supports differential optimization for dynamic Gaussian Splatting. The color $C$ of a pixel $\boldsymbol{u}$ on the image plane is derived using a point-based volume rendering equation by taking samples of density $\sigma$, transmittance $T$, and color $c$ along the ray with interval $\delta_i$ as:

$$C(\boldsymbol{u}) = \sum_{i=1}^{N} T_i \left(1 - \exp(-\sigma_i \delta_i)\right) c_i \quad \text{with} \quad T_i = \exp\left(-\sum_{j=1}^{i-1} \sigma_j \delta_j\right). \tag{2}$$

Adaptive density control manages the density of 3D Gaussians in the optimization, allowing dynamic variation over iterations. For more details, refer to Kerbl et al. (2023).

Dynamic Gaussian Splatting frameworks utilize a canonical space and learn warp fields, denoted as $\mathcal{F}$, which transform canonical Gaussians $\mathcal{G}$ into their deformed states at time $t$. The offset in Gaussian parameters due to this transformation is expressed as:

$$\delta\mathcal{G} = \mathcal{F}_\theta(\mathcal{G}, t). \tag{3}$$

Typically, these deformed geometries are mapped through a neural network Yang et al. (2023); Luiten et al. (2024); Wu et al. (2024), with optional color mapping applied as part of the transformation process. Following deformation, a novel-view image $\hat{I}$ is rendered using differential rasterization with a view matrix $V$ and target time $t$. While MLP-based deformation mapping often suffers from overfitting to training views, leading to degraded novel-view reconstruction, we address this by enforcing analytical constraints on the warp field. This ensures Gaussian motions adhere to

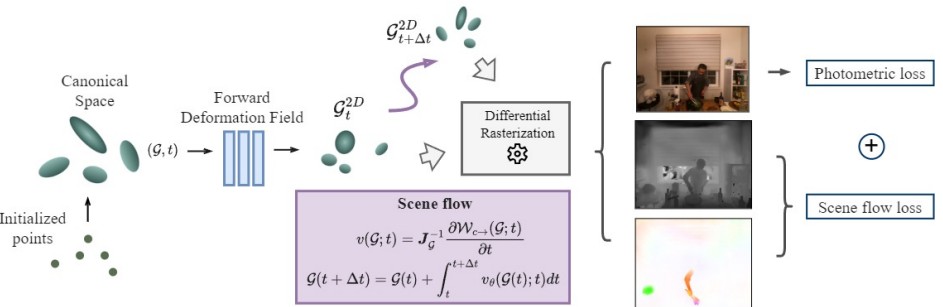

Figure 2: **Analytical scene flow from warp field.** With canonical Gaussians $\mathcal{G}_c$, we transform them forward in time to $\mathcal{G}_t$, then perform time integration from warp field velocities $v(\mathcal{G}; t)$ to derive $\mathcal{G}_{t+\triangle t}$. The Gaussian offsets $\mathcal{G}_{t+\triangle t} - \mathcal{G}_t$ are compared to reference scene flow.

the expected scene flow field, mitigating the issues of overfitting and SfM-initialized point cloud limitations.

**Warp field expression with twist increments.** Warp fields, previously defined for point cloud registration tasks Aoki et al. (2019), can be used to model rigid transformations between sets of Gaussian means. Let $\boldsymbol{\mu}_\tau \in \mathbb{R}^{N_1 \times 3}$ and $\boldsymbol{\mu}_s \in \mathbb{R}^{N_2 \times 3}$ represent the target and source Gaussian means, respectively, where $N_1$ and $N_2$ denote the number of Gaussians in each set. The warp field, which defines the rigid transformation aligning the source Gaussian set to the target set in the special Euclidean group $SE(3)$, can be expressed as $\mathcal{W}(\boldsymbol{\xi}) = \exp\left(\sum_{\mu=1}^{6} \boldsymbol{\xi}_\mu P_\mu\right)$.

Here, $\boldsymbol{\xi} \in \mathbb{R}^6$ are the twist parameters, and $P$ are the generator matrices of the group. The warp field is optimized by minimizing the difference between the transformed source Gaussian means and the target means:

$$\underset{\boldsymbol{\xi}}{\operatorname{argmin}} \left\| \phi(\mathcal{W}(\boldsymbol{\xi}) \cdot \boldsymbol{\mu}_s) - \phi(\boldsymbol{\mu}_\tau) \right\|_2^2, \tag{4}$$

where $\phi : \mathbb{R}^{N \times 3} \to \mathbb{R}^K$ is an encoding function that either explicitly extracts geometric features such as edges and normals or implicitly encodes the Gaussian means into feature vectors. The notation $(\cdot)$ represents the application of the warp field transformation.

This optimization can be solved iteratively given the twist parameter Jacobian matrix $J$, denoting the changes in the warp field concerning twist parameters, that can be further decomposed into a product of warp field gradient and the encoding functional gradient as:

$$J = \frac{(\mathcal{W}^{-1}(\boldsymbol{\xi}) \cdot \boldsymbol{\mu}_\tau)}{\partial \boldsymbol{\xi}^T} \cdot \frac{\partial \phi(\mathcal{W}^{-1}(\boldsymbol{\xi}) \cdot \boldsymbol{\mu}_\tau)}{\partial (\mathcal{W}^{-1}(\boldsymbol{\xi}) \cdot \boldsymbol{\mu}_\tau)^T}. \tag{5}$$

The optimization process iteratively learns the twist increment $\Delta \boldsymbol{\xi}$ that best aligns the encoded source Gaussian means with the target Gaussian means.

## 4 GAUSSIAN SPLATTING WARP FIELD REGULARIZATION BY SCENE FLOW

We investigate the problem of fitting deformable Gaussian Splatting from a monocular video sequence. To initialize the process, we employ structure-from-motion methods to set the canonical Gaussians and camera parameters. For a given set of 3D Gaussians $\mathcal{G}$ at time $t$, the warp field deforms these Gaussians from their canonical positions to their dynamic locations at time $t$. Our goal is to find the optimal network parameters $\theta$ for the forward warp field $\mathcal{W}_{c \to}(\mathcal{G}, t)$. This optimization aims to ensure that both the derived scene flow $\hat{\mathcal{T}}_{t \to t+1}$ and the rasterized image $\hat{\mathcal{I}}_t$ closely match the expected flow field $\mathcal{T}_{t \to t+1}$ and the reference image $\mathcal{I}_t$:

$$\mathcal{L} = \mathcal{L}_{color} + \mathcal{L}_{motion} = |\hat{\mathcal{I}}_t - \mathcal{I}_t| + \left\| \hat{\mathcal{T}}_{t \to t+1} - \mathcal{T}_{t \to t+1} \right\|. \tag{6}$$

The scene flow component can be further divided into optical flow and depth estimations, corresponding to in-plane and out-of-plane motions. Consider a Gaussian mean $\boldsymbol{\mu} = \boldsymbol{\mu}(t)$ moving in the

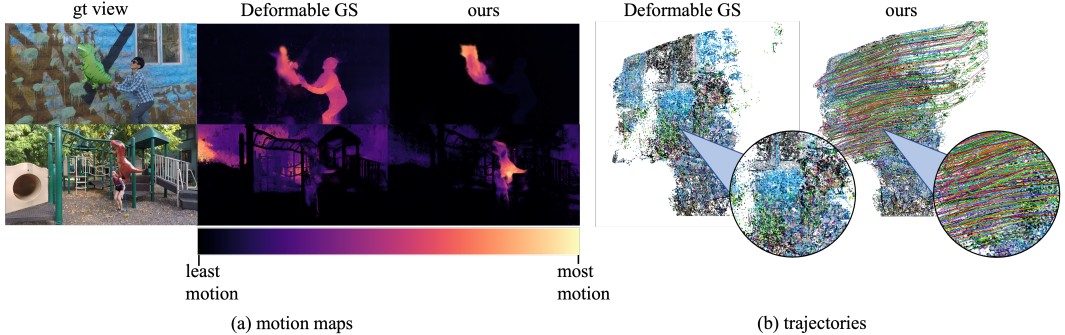

Figure 3: (a) Visualization of Gaussians' travel distance $||\boldsymbol{\mu} - \boldsymbol{\mu}_o||_2$. In both of the scenes, the humans are stationary with dinosaur balloons moving around. Our result correctly identifies the dynamic regions, whereas the baseline model forms motions in the background and on supposedly stationary humans to compensate for photometric correctness. (b) 3D visualization of the motion trajectories. Our result shows clean trajectories from the waving balloon.

scene. Its instantaneous scene flow is $\frac{d\boldsymbol{\mu}}{dt}$. If the corresponding pixel location on the image plane is $\boldsymbol{u}_i = \boldsymbol{u}_i(t)$, the optical flow can be expressed as the projection of the scene flow to the image plane, with $\frac{\partial \boldsymbol{\mu}}{\partial \boldsymbol{u}_i}$ as the instantaneous camera projective relationship:

$$\frac{d\boldsymbol{u}_i}{dt} = \frac{d\boldsymbol{u}_i}{d\boldsymbol{\mu}} \frac{d\boldsymbol{\mu}}{dt}. \tag{7}$$

Deriving an approximation to the scene flow needs the expression reversed. We can represent the relationship between $\boldsymbol{\mu}$ and $\boldsymbol{u}_i$ with dependencies on time as $\boldsymbol{\mu} = \boldsymbol{\mu}(\boldsymbol{u}_i(t); t)$ Vedula et al. (2005). By differentiating this mapping with respect to time, we obtain:

$$\frac{d\boldsymbol{\mu}}{dt} = \frac{\partial \boldsymbol{\mu}}{\partial \boldsymbol{u}_i} \frac{d\boldsymbol{u}_i}{dt} + \frac{d\boldsymbol{\mu}}{dt}\Big|_{\boldsymbol{u}_i}. \tag{8}$$

This equation describes the motion of a point in 3D space while decomposing the motion of a Gaussian in the world into two components. The in-plane component $\frac{d\boldsymbol{u}_i}{dt}$ represents the instantaneous optical flow, while the out-of-plane term reflects the motion of $\boldsymbol{\mu}$ along the ray corresponding to $\boldsymbol{u}_i$. This decoupling allows us to regularize the optical flow field and the depth field separately from the estimated scene flow.

The scene flow regularization can thus be re-expressed as a combination of optical flow loss applied over the interval of $(t \to \Delta t)$ and depth loss of Gaussians at $(t + \Delta t)$, where the next step Gaussians are analytically derived:

$$\mathscr{L}_{motion}(t) = \left\| \hat{\mathcal{T}}_{t \to t+1} - \mathcal{T}_{t \to t+1} \right\| = \alpha \mathscr{L}_{flow} + \beta \mathscr{L}_{depth}. \tag{9}$$

Decoupling the motions alleviates the constraints on the supervision dataset and adds flexibility to the optimization process. A visualization of the regularization pipeline is provided in Fig. 2. Further implementation details of the losses are available in Appendix A.1.

## 4.1 FORWARD WARP FIELD VELOCITY

To address the challenge of transforming canonical Gaussians $\mathcal{G}$ over time, our goal is to derive the velocity field from the forward warp field. Specifically, we seek to identify a warp field $\mathcal{W}$ that advances the canonical Gaussians in time, with an underlying velocity field $v(\mathcal{G}; t)$ that quantifies the changes in Gaussian parameters within the world coordinate system at a specific time $t$. This derivation is conceptually similar to that of twist Jacobians Aoki et al. (2019); Li et al. (2021b) and the analytical warp regularization in NeRF Wang et al. (2023), effectively capturing continuous deformation over time.

In the context of Gaussian Splatting, the motion of Gaussians between two adjacent frames is bijective because all Gaussians are transformed from the same canonical space, creating a one-to-one

correspondence among the Gaussians. Consequently, we can compute the velocity field directly from the forward warp field $\mathcal{W}_{c\rightarrow}(\mathcal{G}, t)$. To analytically derive the velocity field $v(\mathcal{G}; t)$, we must represent the gradient of the warp field in relation to time and warp parameters. Assuming that the Gaussian motions are small, we can utilize a modified Lucas-Kanade expression. By applying a Taylor series expansion to the warp field, we decompose the velocity into two essential components: the feature gradient of the warp field with respect to the corresponding Gaussian parameter and the analytical warp Jacobian Wang et al. (2023); Aoki et al. (2019). This relationship is expressed as:

$$v(\mathcal{G}; t) = \frac{d\mathcal{W}_{c\rightarrow}(\mathcal{G}, t)}{dt}\bigg|_{(\mathcal{G}, t)} = J_{\mathcal{G}}^{-1} \frac{\partial \mathcal{W}_{c\rightarrow}(\mathcal{G}; t)}{\partial t}. \tag{10}$$

In this equation, the term $\frac{\partial \mathcal{W}_{c\rightarrow}(\mathcal{G}; t)}{\partial t}$ indicates how each output Gaussian parameter varies with respect to the input canonical Gaussian parameters. Meanwhile, $J_{\mathcal{G}}$ describes how modifications in the warp field output influence the transformation of the Gaussians. In practical scenarios, the warp Jacobian can face numerical instabilities when $det(J_{\mathcal{G}}) < \epsilon$, especially if the canonical Gaussian scene has not yet stabilized. Discarding these unstable motions can negatively impact subsequent densification and pruning processes. To mitigate these instabilities, we substitute the inverse Jacobian $J_{\mathcal{G}}^{-1}$ with the Moore–Penrose pseudoinverse $J_{\mathcal{G}}^{+}$.

The derived expression is valid for any Gaussian parameter input into the deformation network, provided the input and output dimensions are consistent. However, given that the scaling terms and spherical harmonics of the Gaussians show minimal changes between adjacent frames, we choose to omit these from the calculations for improved efficiency. Summarizing, we compute the velocity fields corresponding to the displacement and rotation of Gaussians as follows:

$$v(\boldsymbol{\mu}; t) = J_{\boldsymbol{\mu}}^{+} \frac{\partial \mathcal{W}_{c\rightarrow}(\boldsymbol{\mu}; t)}{\partial t}; \quad v(\boldsymbol{q}; t) = J_{\boldsymbol{q}}^{+} \frac{\partial \mathcal{W}_{c\rightarrow}(\boldsymbol{q}; t)}{\partial t}, \tag{11}$$

where $v(\boldsymbol{\mu}; t) : \mathbb{R}^3 \times \mathbb{R} \rightarrow \mathbb{R}^3$ and $v(\boldsymbol{q}; t) : \mathbb{R}^4 \times \mathbb{R} \rightarrow \mathbb{R}^4$ describe the positional and rotational velocity of a Gaussian in the world coordinate, respectively.

## 4.2 SCENE FIELD FROM TIME INTEGRATION

**Time integration on the velocity field.** Given the velocity field from equation 11, and 3D Gaussians $\mathcal{G}(t)$ observed at time step $t$, we can apply time integration using a Runge-Kutta Batschelet (1952) numerical solver on the velocity field to obtain the offset parameters for Gaussians at time $(t + \Delta t)$ Specifically, we employ Monte Carlo integration with a sampling size of 10 points over each domain, which provides an efficient and well-balanced alternative to deterministic approaches. The integration is formulated as:

$$\mathcal{G}(t + \Delta t) = \mathcal{G}(t) + \int_{t}^{t+\Delta t} v(\mathcal{G}(t); t)dt. \tag{12}$$

We then train the warp field network, identical to the one outlined in Yang et al. (2023), to minimize the difference between the analytically derived scene flow, and the ground truth scene flow decomposed into optical flow and depth describing the in-plane and out-of-plane Gaussian motions.

**Optical flow and depth rendering from Gaussians.** For each Gaussian at time $t$, we compute each of the corresponding positions and rotations at $(t + \Delta t)$. For rendering the optical flow and depth map, we consider the full freedom of each Gaussian motion.

For rendering the optical flow map $\hat{\mathcal{O}}_{t\rightarrow t+1}$, we want to calculate the motion's influence on each pixel $\boldsymbol{u_i}$ from all Gaussians in the world coordinate. In the original 3D Gaussian Splatting, a pixel's color is the weighted sum of the projected 2D Gaussian's radiance contribution. Analogous to this formulation, the optical flow can be rendered by taking the weighted sum of the 2D Gaussians' contribution to pixel shifts, as derived in previous literature on flow regularization Gao et al. (2024); Guo et al. (2024):

$$\hat{\mathcal{O}}_{t\rightarrow t+1} = \Sigma_{i\in N} o_i \alpha_i \prod_{j=1}^{i-1} (1 - \alpha_j) = \sum_{i=1} w_i [\Sigma_{i, t_2} \Sigma_{i, t_1}^{-1} (\boldsymbol{u}_{t_1} - \boldsymbol{\mu}_{i, t_1}) + \boldsymbol{\mu}_{i, t_2} - \boldsymbol{u}_{t_1}], \tag{13}$$

where $\mathcal{O}_i$ denotes the optical flow of Gaussian $\mathcal{G}_i$ over the time interval, and $w_i = \frac{T_i \alpha_i}{\Sigma_i T_i \alpha_i}$ denotes the weighing factor of each Gaussian from $\alpha$-blending.

Similarly, the frame's z-depth estimates $\hat{D}_{t+\Delta t}$ can be rendered from the discrete volume rendering approximation accounting for the per-Gaussian contributions:

$$\hat{D}_{t+\Delta t} = \Sigma_{i \in N} d_{i,t_2} \alpha_i \prod_{j=1}^{i-1} (1 - \alpha_j), \tag{14}$$

where $d_{i,t_2}$ denotes the z-depth value of analytically derived Gaussian $\mathcal{G}_i$ from the viewing space at $(t + \Delta t)$. We normalize the optical flow and depth estimates for numerical stability. The optical flow and depth maps are rasterized simultaneously in the same forward pass with color. A performance comparison with various flow and depth priors can be found in A.3.

## 5 EXPERIMENTS

In this section, we first provide the implementation details of the proposed warp field regulation and then validate our proposed method on five dynamic scene datasets captured with different levels of camera movements. Our method outperforms baseline approaches in both static and dynamic camera settings, achieving state-of-the-art results in quantitative and qualitative evaluations.

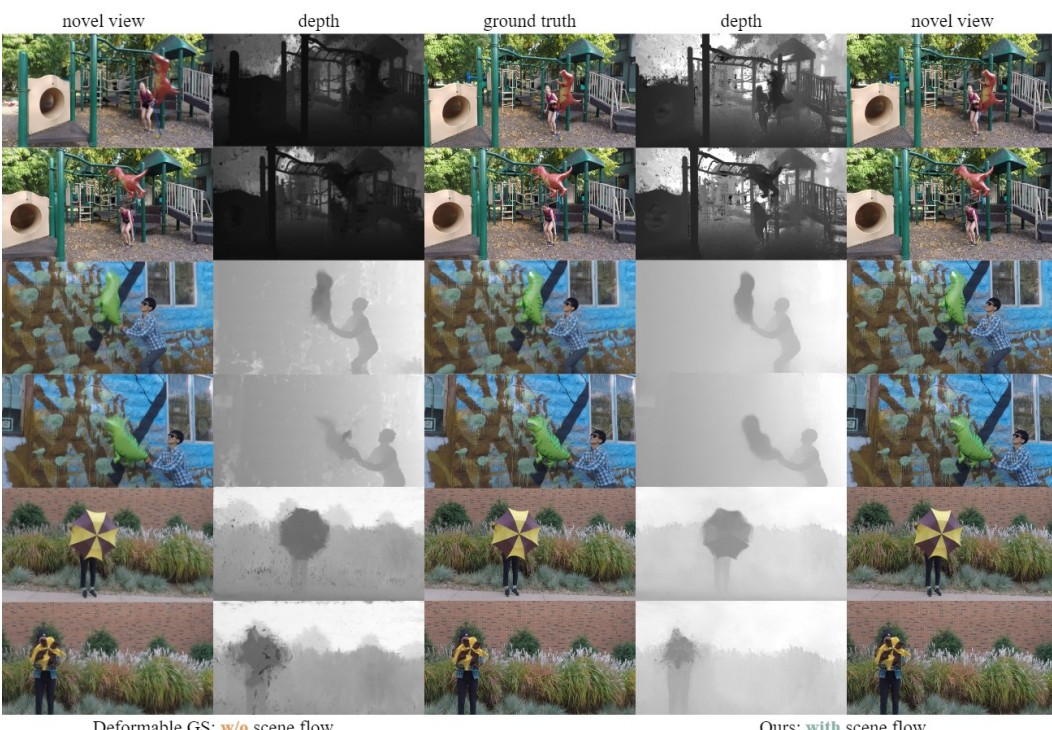

Figure 4: **Qualitative comparisons on the Dynamic Scenes dataset.** Compared to the baseline method, our approach can achieve superior rendering quality on real datasets with lower EMFs.

### 5.1 IMPLEMENTATION DETAILS

We implement the warp field using PyTorch Paszke et al. (2019), leveraging its `Autodiff` library for gradient and Jacobian computations. The framework is optimized with Adam Kingma & Ba (2015) as with 3DGS Kerbl et al. (2023) on a NVIDIA A100 Tensor Core Nvidia Corporation (2024). We use the `Torchdiffeq` Chen (2018) library for numerical integration. Modified from Deformable GS, the inputs to the warp field include 3D Gaussian positions ($\boldsymbol{\mu}$), time ($t$), and 3D Gaussian rotations ($\boldsymbol{q}$) in quaternions. Velocity derivations are enabled by removing the stop gradient operation from the network inputs. Positional encoding Mildenhall et al. (2021a) is applied to extend inputs' frequency band Zheng et al. (2022). We refer readers to Appendix A.2 for more implementation details.

Table 1: **Quantitative evaluation of novel view synthesis on the Dynamic Scenes dataset.** See Sec. 5.2 for descriptions of the baselines.

| method | Playground | | | Balloon1 | | | Balloon2 | | | Umbrella | | |
|---|---|---|---|---|---|---|---|---|---|---|---|---|
| | PSNR ↑ | SSIM ↑ | LPIPS ↓ | PSNR ↑ | SSIM ↑ | LPIPS ↓ | PSNR ↑ | SSIM ↑ | LPIPS ↓ | PSNR ↑ | SSIM ↑ | LPIPS ↓ |
| NSFF | 24.69 | 0.889 | 0.065 | 24.36 | 0.891 | 0.061 | 30.59 | 0.953 | 0.030 | 24.40 | 0.847 | 0.088 |
| NR-NeRF | 14.16 | 0.337 | 0.363 | 15.98 | 0.444 | 0.277 | 20.49 | 0.731 | 0.348 | 20.20 | 0.526 | 0.315 |
| Nerfies | 22.18 | 0.802 | 0.133 | 23.36 | 0.852 | 0.102 | 24.91 | 0.864 | 0.089 | 24.29 | 0.803 | 0.169 |
| (w flow) | 22.39 | 0.812 | 0.109 | 24.36 | 0.865 | 0.107 | 25.82 | 0.899 | 0.081 | 24.25 | 0.813 | 0.123 |
| Flow-sup. NeRF | 16.70 | 0.597 | 0.168 | 19.53 | 0.654 | 0.175 | 20.13 | 0.719 | 0.113 | 18.00 | 0.597 | 0.148 |
| Deformable GS | 24.82 | 0.646 | 0.343 | 22.40 | 0.833 | 0.137 | 24.19 | 0.818 | 0.153 | 22.35 | 0.711 | 0.186 |
| 4DGaussians | 21.39 | **0.776** | 0.204 | 24.48 | **0.849** | 0.144 | 24.72 | 0.801 | 0.219 | 21.29 | 0.560 | 0.332 |
| Ours | **26.34** | 0.756 | **0.184** | **26.35** | 0.848 | **0.133** | 25.89 | **0.911** | 0.151 | **23.02** | **0.746** | **0.176** |

Table 2: **Quantitative results on the DyCheck dataset.**

| Method | Apple | Spin | Block | Teddy | Paper Windmill |
|---|---|---|---|---|---|
| NSFF | 17.54 | 18.38 | 16.61 | 13.65 | 17.34 |
| Marbles | 17.57 | 15.49 | 16.88 | 13.57 | 18.67 |
| 4D Gaussians | 15.41 | 14.41 | 11.28 | 12.36 | 15.60 |
| Ours | 16.03 | 16.71 | 15.46 | 13.60 | 17.41 |
| Ours - pose refined | 22.61 | 24.09 | 23.27 | 17.78 | 19.91 |

Table 3: **Quantitative results for the Deformable GS method.**

| Method | PSNR | SSIM | LPIPS |
|---|---|---|---|
| Deformable GS w/ flow | 20.97 | 0.690 | 0.31 |
| Deformable GS w/ depth | 21.17 | 0.673 | 0.29 |
| Deformable GS w/ both | 23.94 | 0.702 | 0.28 |
| Deformable GS | 24.82 | 0.646 | 0.34 |
| Ours | 26.34 | 0.756 | 0.18 |

## 5.2 EVALUATION

Here we analyze the performance of our method quantitatively and qualitatively. We aim to study if the analytical scene flow regularization on the warp field helps disambiguate the dynamic scene geometry and promote the reconstruction of low EMF scenes.

**Benchmarked datasets.** We evaluate our method using real-world monocular datasets with varying camera motion. These include DyCheck dataset Gao et al. (2022), captured with handheld monocular cameras; Dynamic Scene Yoon et al. (2020), captured by a stationary multi-view 8 camera rig with significant scene motion; Plenoptic Video Li et al. (2021c), captured using a static rig with 21 GoPro cameras GoPro (2023); Hypernerf Park et al. (2021b), which captures objects with moving topologies from a moving camera, suitable for quasi-static reconstruction; and sequences from DAVIS 2017 Perazzi et al. (2016) dataset containing near-static monocular videos.

**Quantitative baseline comparisons.** We evaluate our method's novel-view synthesis capabilities on Plenoptic Videos, Hypernerf, Dynamic Scenes, and DyCheck, ordered by decreasing Effective Multi-view Factor (EMF) Gao et al. (2022). These datasets present increasing challenges, with DyCheck posing the greatest difficulty due to its low EMF and unreliable object motion. We first assess our method on Hypernerf and Plenoptics with higher EMFs against recent methods 3DGS Kerbl et al. (2023), Deformable GS Yang et al. (2023), 4DGaussians Wu et al. (2024), and MA-GS Guo et al. (2024). We report the standard metrics LPIPS Zhang et al. (2018), SSIM Wang et al. (2004), and PSNR Welstead (2009) in Table 5 and 4. We see that our method consistently outperforms these Gaussian baselines on scenes with higher EMFs.

To further assess our method's performance on monocular sequences with lower EMFs, we evaluate its performance against recent methods Marbles Stearns et al. (2024), NSFF Li et al. (2021d), NR-NeRF Tretschk et al. (2021b), Nerfies Park et al. (2021a), and Flow-supervised NeRF **?**, see Table 1 and 2. These comparisons highlight the effectiveness of our analytical scene flow approach compared to other flow-guided rendering frameworks. Notably, our method outperforms NSFF, which uses a similar supervision style but a different scene flow derivation. The comparison with Flow-supervised NeRF, which also uses analytical scene flow derived from an implicit radiance field, serves as an ablation study, demonstrating the effectiveness of our derived scene flow on an explicit scene representation. A.3.

**Comparisons with direct depth and flow Supervision.** We note that it is essential to understand the performance improvements over regularizing the deformable network directly with depth and flow priors. We include ablation experiments with Deformable GS supervised with depth and flow losses, see Table 3. The results suggest that directly applying depth and flow regularizations encourages the scene to overfit to training viewpoints, leading to degraded performance. With depth regularizations, the scene geometries become more prominent, with a sacrifice in motion smoothness. Regularizing offsets between two frames can enforce accurate piecewise motion but still cannot generalize well to

Table 4: **Quantitative evaluation of novel view synthesis on the Plenoptic Videos dataset.** See Sec 5.2 for an analysis of the performance.

| method | cook spinach | | | cut roasted beef | | | sear steak | | | Mean | | |
|---|---|---|---|---|---|---|---|---|---|---|---|---|
| | PSNR ↑ | SSIM ↑ | LPIPS ↓ | PSNR ↑ | SSIM ↑ | LPIPS ↓ | PSNR ↑ | SSIM ↑ | LPIPS ↓ | PSNR ↑ | SSIM ↑ | LPIPS ↓ |
| Deformable GS | 32.97 | 0.947 | 0.087 | 30.72 | 0.941 | 0.090 | 33.68 | 0.955 | 0.079 | 32.46 | 0.948 | 0.085 |
| 4DGaussians | 31.98 | 0.938 | **0.056** | 31.56 | 0.939 | **0.062** | 31.20 | 0.949 | **0.045** | 31.58 | 0.942 | **0.055** |
| MA-GS | 32.10 | 0.937 | 0.060 | 32.56 | 0.941 | 0.059 | 31.60 | 0.951 | 0.045 | 32.09 | 0.943 | 0.053 |
| Ours | **33.91** | **0.951** | 0.064 | **32.40** | **0.957** | 0.084 | **34.02** | **0.963** | 0.057 | **33.44** | **0.954** | 0.068 |

Table 5: **Quantitative evaluation on the Hypernerf dataset.** See Sec. 5.2 for detailed explanations.

| method | PSNR ↑ | SSIM ↑ | LPIPS ↓ |
|---|---|---|---|
| 3DGS | 20.84 | 0.70 | 0.45 |
| Deformable GS | 26.47 | 0.79 | 0.29 |
| 4DGaussians | 26.98 | 0.78 | 0.31 |
| Ours | **27.38** | **0.81** | **0.26** |

Table 6: **Ablation study on scene flow decomposition on "Playground" scene from Dynamic Scenes dataset.** See Sec. 5.2 for detailed descriptions of each ablated design.

| method | PSNR ↑ | SSIM ↑ | LPIPS ↓ |
|---|---|---|---|
| w/ Depth Sup. | 24.76 | 0.711 | 0.274 |
| w/ Flow Sup. | 24.37 | 0.695 | 0.229 |
| Ours | 26.34 | 0.756 | 0.184 |

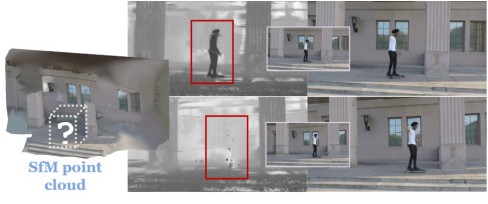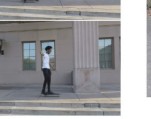
**SfM point cloud**

(a) Unstable reconstruction when SfM fails to reconstruct dynamic region.

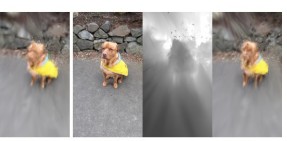

(b) Inaccurate camera poses lead to degraded rendering.

Figure 5: Our method struggles with insufficient point cloud initialization due to reliance on non-rigid warping of scene geometry. In the "skating" scene, this results in unstable geometry at certain angles. Inaccurate camera calibrations also degrade our method, as shown in the "toby-sit" sequence, where miscalibration distorts the scene geometry.

the discontinued unseen viewpoints. In Table 6, we ablate the analytical regularization and report the outcomes of the scene. As shown, each of the scene flow components helps in achieving high-quality synthesis. These ablation experiments reaffirm the validity of the proposed warp field regularization.

**Trajectories from scene flow fields.** We visualize the scene flow trajectories from Dynamic Scenes Dataset sequences to assess the quality and smoothness of the derived flow field. The Gaussians are subsampled from the dynamic regions of the canonical Gaussian space, and then time integrated to produce their respective displacements across time, visualized as colored trajectories in Fig. 3. We note that the trajectories are smooth and follow the expected motions of the dynamic objects. The visualized trajectories of the sequences demonstrate the framework's potential to be extended for tracking in dynamic 3D Gaussian scenes. More trajectories on the DAVIS dataset will be shown in Appendix Fig. A7.

**Geometric consistency.** Fig. 4 compares our method with Deformable GS, which learns Gaussians' time dependency without scene flow regularization. While Deformable GS renders visually accurate novel views, the underlying scene structures are inaccurate, leading to blurry dynamic objects and artifacts in depth maps. Our method, leveraging analytical scene flow regularization, achieves more accurate geometries, reflected in both visual and quantitative results (Table 1). This also enables accurate Gaussian motion tracking (Fig. 3). To explore alternative design choices, we compare with the direct flow and depth supervision on Deformable GS (Fig. 1). These comparisons indicate that our approach, which enforces regularization on the expected scene flow over a continuous time frame, results in more plausible scene structures compared to discrete time step supervision. We additionally conduct experiments on the DyCheck benchmark dataset, see Table 2. The additional visual comparison is shown in Appendix Fig. A2 with comparisons to the deformable baseline. Our results show that the analytical regularization significantly improves the render quality of the DyCheck dataset, especially when accurate camera poses are provided Wang et al. (2024).

**Static and dynamic motion separation.** Our approach formulates deformations within a canonical Gaussian space. To assess the fidelity of our method in separating static and dynamic regions, we visualize the travel distance of each Gaussian from its canonical projection to a queried viewing camera in Fig. 3. The plots are color-coded by the absolute travel distance, with yellow indicating larger distances and purple indicating smaller distances.

In the Playground scene featuring human-object interactions, the visualized travel distance is greatest at the far-side boundaries of the moving object as it recedes from the human. The scene's background is correctly rendered as static with minimal motion. This visualization demonstrates the effectiveness of our method in accurately identifying dynamic regions. Moreover, these results suggest promising future directions for optimizing dynamic scene rendering by filtering out static regions from the optimization process, potentially reducing computational costs.

## 6 LIMITATIONS

**Sensitivity to SfM Initialization:** Our approach is sensitive to the quality of the Structure-from-Motion (SfM) Gaussian space initialization. Accurate initialization is crucial for 3D Gaussian Splatting, particularly without other scene priors. As shown in Fig. 5.a, reconstruction quality deteriorates when dynamic regions have insufficient initialized points.

**Camera Parameter Sensitivity:** Our method is sensitive to inaccuracies in bundle-adjusted camera intrinsic and extrinsic parameters. Errors in these parameters lead to inaccurate rasterization projections, causing noisy geometry or failed reconstruction, as shown in Fig. 5.b.

**Computational Complexity:** The pseudo-inverse operation for velocity field computation requires $(N \times d)$ square matrix decompositions, where $N$ is the number of Gaussians, and $d$ is the number of Gaussian parameters. Potential mitigations include optimized implementations or pre-filtering to remove static Gaussians.

## 7 CONCLUSION

This paper presents a method for incorporating scene flow regularization into deformable Gaussian Splatting. We derive an analytical scene flow representation, drawing upon the theoretical foundations of rigid transformation warp fields in point cloud registration. Our approach significantly enhances the structural fidelity of the underlying dynamic Gaussian geometry, enabling reconstructions of scenes with rapid motions.

Comparison with other Deformable Gaussian Splatting variants demonstrates that regularizing on the warp-field derived scene flow produces more accurate dynamic object geometries and improved motion separation due to its continuous-time regularization. Additionally, comparison with flow-supervised NeRFs reveals the advantages of using Gaussians' explicit scene representations.

While our method exhibits limitations as discussed in the previous section, the accurate geometries learned from dynamic scenes open up promising possibilities for future applications, including 3D tracking from casual hand-held device captures and in-the-wild dynamic scene rendering.

## ACKNOWLEDGEMENT

This work was supported by Fujitsu.

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

## A    APPENDIX / SUPPLEMENTAL MATERIAL

### A.1    SCENE FLOW REGULARIZATION FORMULATION

Recall that in addition to the photometric regularization as detailed in the original 3D Gaussian Splatting Kerbl et al. (2023), $\mathcal{L}_{motion}$ is used as the structural regularization for the derived scene flow consisting of weighted depth and optical flow terms $\mathcal{L}_{motion}(t) = \alpha \mathcal{L}_{flow} + \beta \mathcal{L}_{depth}$.

The optical flow term minimizes the absolute error between the rendered optical flow map as computed from accounting for all Gaussians' 2D projected motion contributions to each rendered pixel, and the reference flow from RAFT Teed & Deng (2020). In particular, the optical flow term is expressed as:

$$\mathcal{L}_{flow} = |\mathcal{O}_{t \to t+1} - \hat{\mathcal{O}}_{t \to t+1}|. \tag{15}$$

As noted in SparseNeRF Guangcong et al. (2023), monocular depth maps are scale-ambiguous thus directly using monocular depth priors with $p$-norm regularizations hinders the spatial coherency. We apply a local depth ranking loss on the rendered depth maps. We transfer the depth ranking knowledge from monocular depth $d_{mono}$ to rendered $d_r$ with:

$$\mathcal{L}_{depth} = \sum_{d_{mono}^{k_1} < d_{mono}^{k_2}} max(d_r^{k_1} - d_r^{k_2} + m, 0), \tag{16}$$

where $m$ is a tunable small margin defining the permissible depth ranking errors, and $k_1, k_2$ are the indices of the randomly queried pixels. If the depth ranking of the monocular depth pixels and the rendered pixels are not consistent, then it entails inaccuracies in the warped Gaussians at time $(t + \Delta t)$, penalizing the learned warp field.

The coefficients $\alpha$ and $\beta$ reweight the magnitude of loss terms to compensate for regularizing with monocular depth cues. In practice, we notice that setting the two coefficients with similar magnitude yields satisfactory results for scenes initialized with dense point clouds. Whereas with insufficient SfM initialization, the motion coefficient $\alpha$ needs to be weighed down in earlier iterations to allow the canonical scene to stabilize.

### A.2    ADDITIONAL IMPLEMENTATION DETAILS

**Scene initialization.** We define the initialization stage as the first $N$ iterations without deploying scene flow regularizations. To warm up the optimization, during the initialization stage, we only compute the photometric losses with the warp field network unfrozen. Note that we discarded the static scene initialization stage typically adopted by other dynamic Gaussian splatting frameworks as it provided incremental benefits.

The length of the warm-up period is defined based on the quality of the SfM scene. For scenes with rich point clouds to start with, we employ 3000 iterations for initialization to refine the canonical structures. Some highly dynamic scenes contain only a few points or no points in the dynamic regions. Since our method relies on analytical warping from the canonical scene and needs Gaussians at the dynamic regions to start with, we extend the warm-up period to $5000$-$10,000$ to recover as many Gaussians as possible at the dynamic regions.

**Motion masking.** The initialized Gaussians generally carry inaccurate motions to compensate for photometric accuracies, regardless of whether the Gaussians are located at the presumably static or dynamic regions. Thus we first randomly sample pixels from the rendering to correct motions of all Gaussians equally across the scene. Since static regions are usually larger than dynamic regions and are easier to converge, this allows us to disambiguate the static and dynamic regions.

After ensuring that most Gaussians in the static regions have been sufficiently constrained, we apply a large motion mask to reconstruct small and fast-moving objects. The motion mask is defined similarly as prior work Li et al. (2021d) as a binary segmentation mask at pixels with normalized motions larger than 0.1. This additional motion masking step allows us to focus on rapid motions when most of the scene motions have converged.

**Training details.** The hyperparameters for learning the Gaussian parameters are kept consistent as the original 3D Gaussian Splatting Kerbl et al. (2023). For comparable analysis on the analytical

warp field regularization, we follow setups in the baselines Yang et al. (2023); Wu et al. (2024). The learning rate of the warp field network is empirically set to decay from $8e$-4 to $1.6e$-6, and Adam's $\beta$ range is set to $(0.9, 0.999)$. As with data preparation, we used COLMAP Schönberger & Frahm (2016) and Reality Capture Epic Games (2024) to estimate camera intrinsics and extrinsics, which are kept fixed during optimization. We use Marigold Ke et al. (2024) and RAFT Teed & Deng (2020) for depth and flow maps.

### A.3  FURTHER DISCUSSIONS ON MOTION LEARNING

**Geometry enhances novel view reconstruction quality.** Fig. A1 presents novel view synthesis results and learned geometries from our method alongside two baseline models. This comparison highlights the impact of accurate underlying geometries on synthesized novel views. All three models converge on the training views, so we focus on the results for unseen viewpoints.

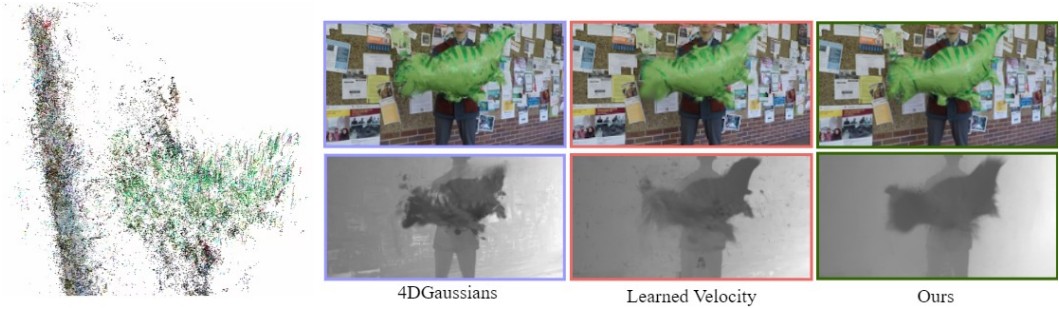

Figure A1: Visualization of Gaussian trajectories and comparison of novel view synthesis results with learned geometries.

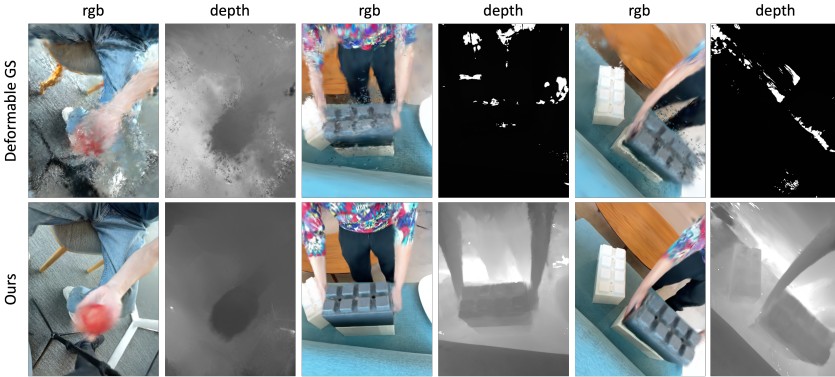

Figure A2: Qualitative comparison of results on scenes from the DyCheck dataset.

4DGaussians Wu et al. (2024) learns the warp field without incorporating motion cues. This approach leads to a noisy underlying geometry with artifacts around specular areas, likely introduced to compensate for photometric inaccuracies. Consequently, its synthesized dinosaur exhibits blurriness around the edges. Another baseline predicts velocities directly instead of relying on analytical derivation as in Du et al. Du et al. (2021). We adopt the velocity field model from Du et al. and perform time integration to derive scene flow. However, this data-driven velocity field tends to produce an "averaged" geometry, resulting in blurriness around the dinosaur's head area.

In contrast, our method generates the most crisp depth map and the highest quality synthesized result. This improvement stems from our ability to accurately learn dynamic geometries. The left-hand side of the figure demonstrates that even with monocular sequences, our method effectively separates the foreground, background, and carried object. Our method serves as a general tool for enhancing dynamic Gaussian splatting frameworks without requiring additional network modules. This is further exemplified by the performance on DyCheck dataset, as shown in Fig. A2

**Motion thresholding visualizations.** Fig. A3 presents a comparison of motion thresholds at different travel distances and our learned trajectories. The first row visualizes the 3D trajectories of Gaussians, color-coded by z-depth magnitude and in learned color. This visualization clearly demonstrates the accurate motions of a human waving a balloon. The second row displays the motion plot generated by our method, while the third row showcases the results from the DeformableGS baseline model Yang et al. (2023). The baseline model exhibits a significant error, moving the stationary human instead of the balloon. Notably, the side of the balloon furthest from the human experiences the least movement, while the human's torso exhibits the most displacement. This outcome directly contradicts expectations. In contrast, our method accurately captures the motions, correctly ranking their magnitudes. This highlights the effectiveness of our approach in distinguishing and representing the dynamic elements within the scene.

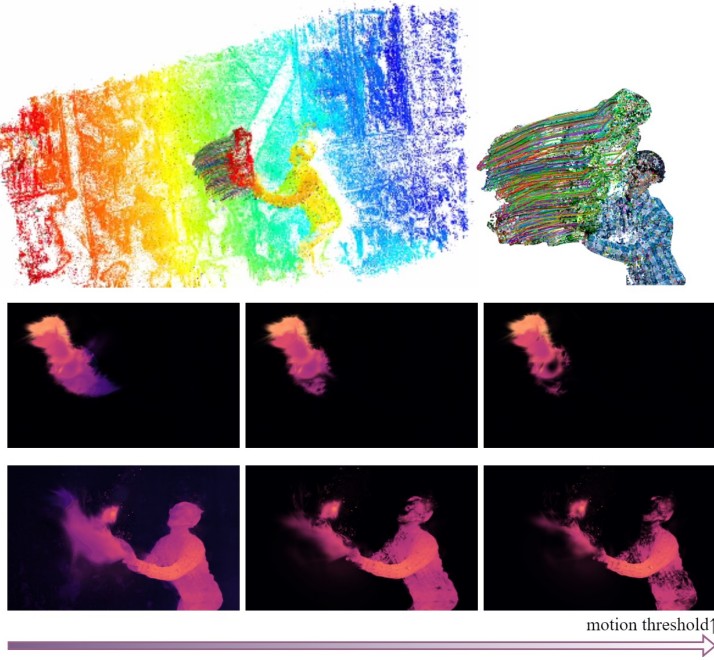

Figure A3: 3D trajectories and thresholded motion plots. The first row of motion plots are results from our method and the second row are results from baseline model DeformableGS.

**Sensitivities to SfM initialization continued.** As discussed in the limitations section, our method relies on analytical warping from canonical space, making it sensitive to Structure-from-Motion (SfM) initialization, particularly when dynamic regions lack sufficient point correspondences. Fig. A4 illustrates this sensitivity using the "train" scene from the DAVIS dataset Perazzi et al. (2016). The left side of the figure shows the SfM-initialized mesh, where the toy train traveling on the circular track is not captured. The top right figure displays the learned trajectories from the baseline model, exhibiting irregular Gaussian movements within the dynamic area.

The bottom right figure presents the learned trajectories from our method. While our method captures the expected trajectory of the moving train, it struggles to reconstruct dense Gaussians for the train due to an insufficient number of initialized Gaussians in this region. Additionally, the Gaussians are not accurately placed at the correct depth, as the monocular depth ranking loss only enforces the correct depth ranking order. Employing alternative depth regularization techniques could improve the learned depth in absolute scale.

**Sensitivity to the choice of depth and flow supervision** Understanding the sensitivity of the choice of depth and flow supervision is crucial for accurately demonstrating the merits of our work. To explore this further, we performed experiments on the "Playground" scene in which different methods for depth and flow supervision were used. Visual comparisons of the ablated priors are shown in Fig. A5. For depth supervision, we tested Depth Anything V2 Yang et al. (2024) and Midas Birkl et al. (2023), achieving PSNR scores of 26.96 and 26.78, respectively. For flow supervision, we

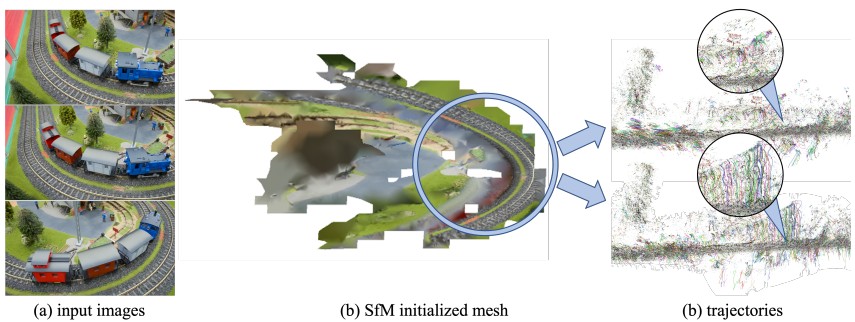

(a) input images         (b) SfM initialized mesh        (b) trajectories

Figure A4: SfM initialized mesh and the reconstructed toy train trajectories. The right top subfigure shows trajectories from the baseline model, and the right bottom subfigure shows trajectories from our model.

evaluated FlowNet Ilg et al. (2017) and MemFlow Dong & Fu (2024), achieving PSNR scores of 26.38 and 27.04. As a reference, our original setup achieved a PSNR of 26.34. The results indicate that Midas yields the poorest performance, producing blurry depth maps. In contrast, Depth Anything V2 provides sharper details but struggles with background accuracy in outdoor scenes. Our chosen method, Marigold Ke et al. (2024), gives more visually accurate depth maps, capturing even distant background details. However, it performs slightly worse quantitatively compared to the others. Nevertheless, the variation in depth priors appears to have minimal impact on our method's final results, as most recent methods exhibit similar performance levels. Regarding flow estimation, FlowNet produced less accurate flow maps, failing to capture scene movement effectively. Conversely, MemFlow, the current state-of-the-art optical flow estimator, achieved the best results with sharp details and accurate flow. Our selected method, RAFT Teed & Deng (2020), also provides accurate flow maps but is not as sharp as MemFlow, which is likely responsible for the improved PSNR.

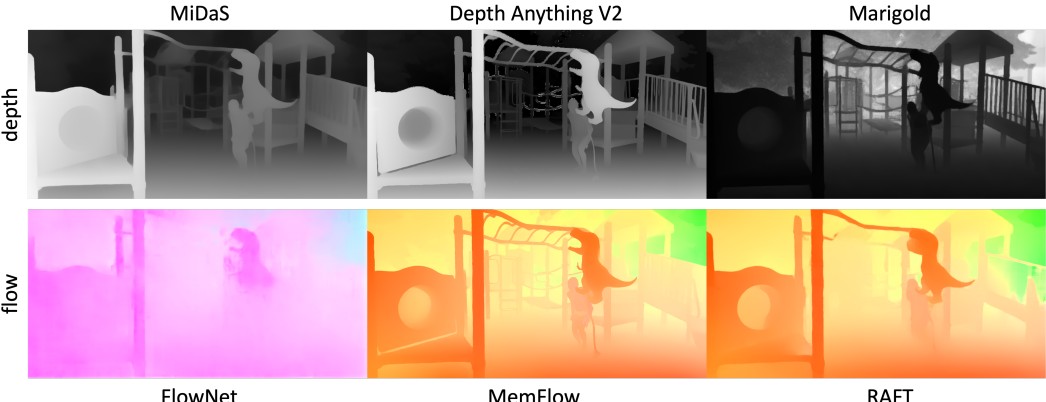

Figure A5: Qualitative comparison of various depth and flow methods applied to the Playground scene. Methods discussed in the paper are shown in the **right** column.

**Network Jacobians as uncertainty measure.** Deformation networks exhibit distinct patterns in regions that are easier to reconstruct compared to those that are more challenging. To explore this, we analyze the Jacobian of these deformations with respect to time, enabling us to quantify uncertainty directly from these patterns. Here in Fig. A6 we present preliminary results on a controlled synthetic scene, where the red and green balls have randomized positions in the training dataset, while the blue ball accurately corresponds to the correct time step. Brighter colors indicate higher amounts of uncertainty.

**More results on real scenes.** More qualitative results on dynamic scenes can be found in Fig. A7.

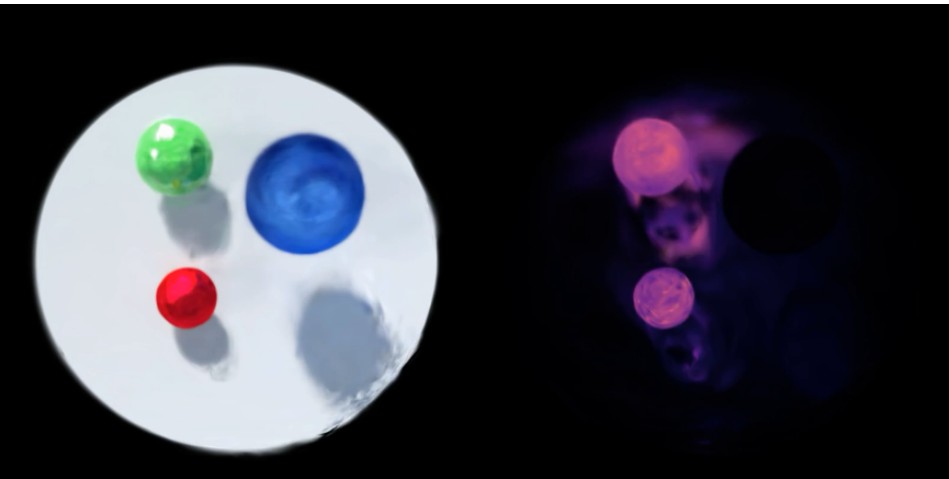

Figure A6: The uncertainties of the Gaussians are visualized in the plot, with the brighter regions higher in magnitude.

.

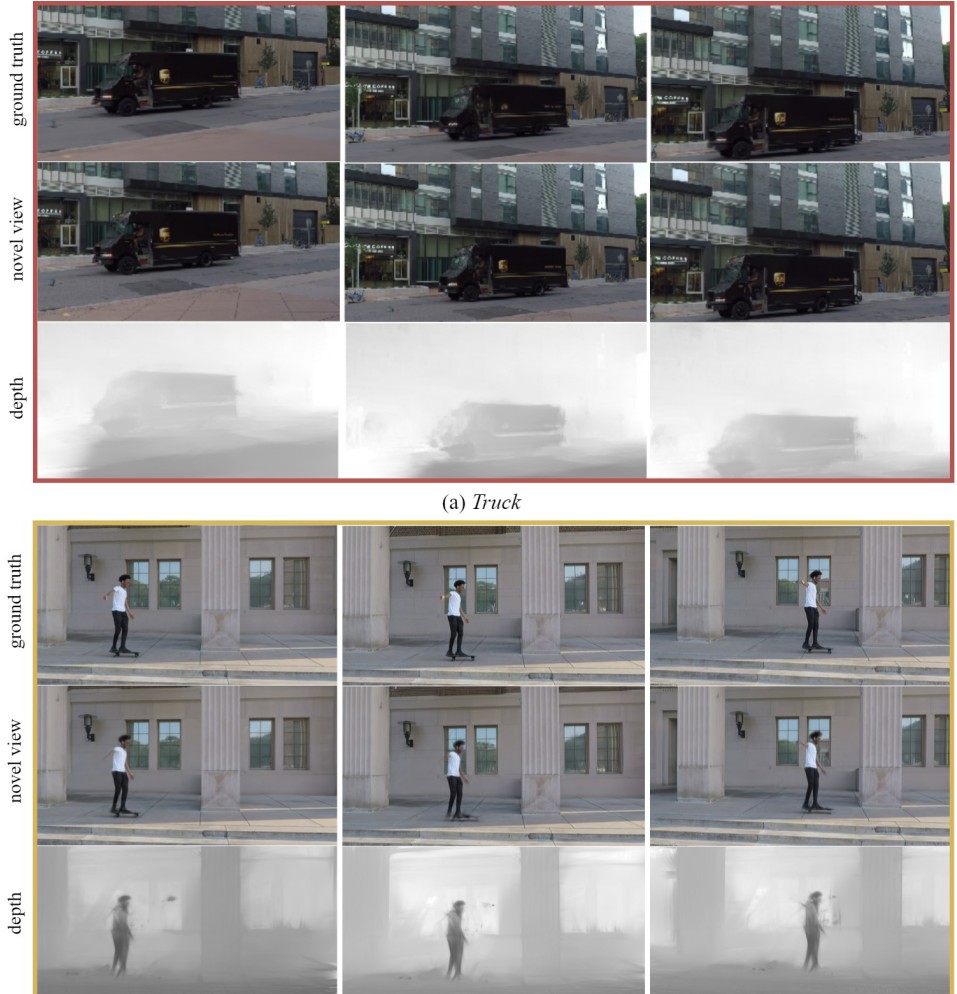

(a) *Truck*

(b) *Skating*

Figure A7: Qualitative comparisons on Dynamic Scenes Dataset Yoon et al. (2020)
.

