# OpenReview forum: "Gaussian Splatting Lucas-Kanade"
_ICLR.cc/2025/Conference — ICLR 2025 Poster_

### Official Review · Reviewer_D8fp · 2024-10-18

**Soundness:** 3
**Presentation:** 3
**Contribution:** 2
**Rating:** 5
**Confidence:** 4

**Summary:**

This paper tackles the challenges posed by dynamics and the lack of motion parallax that Gaussian Splatting methods face in real-world settings. The author proposes an analytical solution that adapts Lucas-Kanade method to dynamic Gaussian splitting. The proposed method leverages the forward warp field and derives an analytical velocity field for accurate scene flow computation. The experiment shows that “GAUSSIAN SPLATTING LUCAS-KANADE” outperforms state-of-the-art methods on both synthetic and real-world scenes, and demonstrates accurate reconstructing capability in complex scenes with minimal camera movement.

**Strengths:**

The proposed method analytically derives the expression of warp field velocities and provides better tractability compared to directly supervising the flow field as the difference in estimated deformations between consecutive frames. Besides, the method is applicable to forward warp field techniques, and ensures accurate Gaussian trajectories throughout the sequence.
In real-world application, the author demonstrates that the warp field regularization accommodates deformations in complex scenes with minimal camera movement, achieving results.
The paper is well-written, concise, and has excellent formatting of figures and formulas.

**Weaknesses:**

Despite the authors' efforts to derive an analytical velocity field and incorporate flow and depth supervision to regularize the Gaussian optimization, I have identified several questions that require attention.

1. The author suggests that the velocity field can be derived from the forward warp field by utilizing a modified Lucas-Kanade expression, and computing the fields corresponding to the displacement and rotation of Gaussians. Nevertheless, I argue that this claim is not entirely accurate. A crucial oversight in the authors' derivation is that optical flow or warp fields are inherently tied to both camera pose and Gaussian motion. Yet, the analysis seems to only consider the scenario flow resulting from the motion of the Gaussian alone, as evident in equations[8][11]. The motivation behind the authors' decision is unclear, but it is likely that incorporating camera pose variations into the Jacobian expression would lead to improved outcomes.

1. The authors should provide a more comprehensive explanation of their implementation, as the description of time integration on the velocity field in Section 4.2 remains unclear. Specifically, what is the value of the time step ∆t employed by the authors, and how is it determined? Furthermore, does the method still satisfy the small motions assumption stated in Line 273 when ∆t is large?

1. The proposed algorithm's reliance on computationally intensive operations, such as the (N × d) pseudo-inverse and time integration, raises concerns about its efficiency. To facilitate a more comprehensive assessment of the method's practical viability, it would be beneficial for the authors to provide a detailed analysis of the time complexity for each algorithmic component, as well as the overall memory requirements. This is essential for evaluating the algorithm's real-world applicability.

1. In my opinion as a reader, the paper's formulas are somewhat verbose, as the Autodiff library is leveraged for efficient gradient and Jacobian computations, obviating the need for manual CUDA implementation. Consequently, it may not be essential to include the detailed derivation process in the main body of the paper.

1. Although the paper's experimental design is thorough, it still exhibits some limitations. The experimental setup is somewhat brief and insufficient. I believe that relevant methods, such as [1][2], should be included as baselines to more thoroughly validate the effectiveness of this method. Additionally, I highly recommend that the authors incorporate a comparison with PhotoSLAM to further bolster the experimental setup.

[1] Motion-aware 3D Gaussian Splatting for Efficient Dynamic Scene Reconstruction

[2] DynMF: Neural Motion Factorization for Real-time Dynamic View Synthesis with 3D Gaussian Splatting

**Questions:**

1. The author should provide more details about the experimental setup, e.g., time step ∆t  in Eq.12. Please clarify how is it determined? does the method still satisfy the small motions assumption stated in Line 273 when ∆t is large?

2. The authors highlight in Section 6 that their method is sensitive to SfM initialization. Nevertheless, in my experience, 3DGS can still produce high-quality renderings even when initialized with random point clouds. I find it puzzling that the authors emphasize this aspect, and I would appreciate it if they could provide a more in-depth explanation for this claim.

3. It would be beneficial if the authors could furnish a detailed analysis of the time complexity for each component of the algorithm, as well as the memory requirements during the training process. This information is essential for assessing the algorithm's efficiency and scalability.

Currently, I would rate this paper as borderline. However, I am willing to reconsider my evaluation if the authors can provide additional clarifications and engage in a more in-depth discussion to address the concerns I have raised.

---

> ### Author Response · Authors · 2024-11-18
> **Responses to Reviewer D8fp**
>
> We thank the reviewer for the positive comments on the paper presentation, writing, and appreciate the thorough constructive feedback! We address the concerns in detail in the following and look forward to engaging in more in-depth discussions.
>
> **The claim that velocity fields can be derived from the forward warp field using modified Lucas-Kanade and displacement/rotation fields is seen as inaccurate; camera pose should also influence flow derivations.**
>
> The velocity field and scene flow field are both derived in a 3D space. The velocities of the Gaussians express their traveling speed in space, regardless of the camera viewpoints or their motions. So the velocity field derived should not be concerned with camera motion influences. Once the scene motion is projected to a certain camera plane, the flattened optical flow on the image plane would be comprised of both scene motions and camera motions, leading to the inclusion of the depth term to constrain the regularization of the scene motions. To this end, as the reviewer points out, it would likely be beneficial to further incorporate the variations in camera pose. We currently only consider the projected scene motion in this work due to scope limitations, but have been considering further disentangling the camera motion's contribution alongside the uncertainties arising from scene and camera motions in the follow-up work.
>
> **The method’s time integration is not well explained (e.g., time step Δt) in Section 4.2.**
>
> The time integration is performed with Monte Carlo integration, with a sampling size of 10 points over each domain. Monte Carlo integration is preferred over other deterministic approaches for efficiency and balanced performance. We can include this detail in the revision if it improves understanding.
>
> **Experimental limitations are noted, and additional baseline comparisons (e.g., with PhotoSLAM) are suggested.**
>
> We have thoroughly compared our approach with relevant papers on dynamic Gaussian scene reconstruction, as well as the ones employing flow supervision. The comparisons on DyCheck, Dynamic Scenes, Plenoptic Videos, and HyperNeRF datasets are reported in Tables 2, 1, 4, and 5, ordered by an increasing Effective Motion Factor in the datasets. We have carefully reviewed the suggested PhotoSLAM paper for its feasibility for comparison, and recognize its merits as a SLAM algorithm with Gaussians. Unfortunately, the paper is proposed for Gaussian SLAM in static scenes, whereas our objective does not cover localizing cameras and rather focuses on rendering dynamic scenes. For the planned extension of this current work, where the camera motion contributions would be more thoroughly studied, PhotoSLAM and this line of work would be incorporated for comparison.
>
> **Could the authors provide details on the experimental setup, particularly the time step Δt in Eq. (12)? Does the method maintain the small motions assumption as Δt increases?**
>
> Each time step is taken as the time difference between two frames. We use the inherent ordering of frames in each dataset without augmenting the frame selection paradigm to show the approach’s flexibility. We do add a randomly sampled Gaussian noise to the interval into (Δt+ϵ) to simulate perturbations for improved robustness. The small motion assumption would hold only when the trajectory is locally 2-Lipschitz, whereas increasing the Δt would break this condition since the trajectories would become piecewise and discontinuous in time.
>
> **The paper claims sensitivity to SfM initialization, yet 3DGS performs well with random point clouds. Can this claim be further clarified?**
>
> We acknowledge that using random initialization for 3DGS can yield reasonable performance on simple, static scenes. However, in more complex and dynamic scenarios, an accurate SfM initialization proves to be advantageous. A recent study (https://theialab.github.io/nerf-3dgs/) evaluated various initialization methods for static 3DGS, providing further evidence that robust initialization often results in improved performance.
>
> **A breakdown of time complexity and memory requirements during training would help evaluate efficiency and scalability.**
>
> We profiled the time consumption and memory requirements on the “Playground” video from the Dynamic Scenes dataset. Further optimizations to inversion implementation such as using Least Squares to approximate inverse instead of employing an exact solution could further reduce the profiled complexities.
> | Method| Memory | Speed |
> | -------- | ------- | ------- |
> | Deformable GS | 10.6 GB | 14.56 it/s |
> | w/ analytical flow (100 Gaussians)| 11.3 GB  | 10.82 it/s |
> | w/ analytical flow (1000 Gaussians) | 12.7 GB  | 6.47 it/s |

---

> > ### Comment · Reviewer_D8fp · 2024-11-25
> > **Clarifying the neglect of camera motion**
> >
> > From my perspective, the so-called sceneflow loss is intended to minimize the distance between the optical flow and the derived sceneflow. However, the optical flow is a composite of camera motion and Gaussian scene motion, whereas the derived term only considers the dynamic Gaussian component. Crucially, camera motion, particularly rotation, is a major contributor to optical flow, the neglect of camera motion is clearly inaccurate.

---

> ### Author Response · Authors · 2024-11-25
>
> Hi, seems like there is a certain level of miscommunication here. When we project the scene flow to a 2D camera plane to obtain optical flow $\hat{O}_{t\rightarrow t+1}$, the rendered optical flow does capture both the camera motion and the scene motion. Since the optical flow derivation and disentanglement of optical flow into camera + motion components are not the focus of our paper, we follow the standard experimental setup in Neural Scene Flow Fields (https://arxiv.org/pdf/2011.13084) only to show the effectiveness of the analytical flow field. With a pixel $p_t$ at time $t$, the pixel's forward location is computed by performing perspective projection of the point's 3D location with its corresponding scene flow displacement into the viewpoint at time $t+1$. This way, it yields a projected scene with both camera and scene motion warped forward in time.
>
> If the camera motion was not considered in this setup, as suggested here, then the NVS result would have been blurry due to a lack of camera motion regularization, and the same consequence would have happened with many prior NeRF/GS papers following the same experimental setup. While an additional explicit regularization term for camera motion might be beneficial, this is not the main consideration of our paper.

---

> > ### Author Response · Authors · 2024-12-01
> >
> > Thank you for your thoughtful feedback. We’ve addressed your concerns in the rebuttal, and welcome further discussions. If no issues remain, we kindly ask you to consider updating your score to reflect the clarifications. We appreciate your time and consideration!

---

### Official Review · Reviewer_uwP8 · 2024-11-04

**Soundness:** 2
**Presentation:** 2
**Contribution:** 2
**Rating:** 6
**Confidence:** 4

**Summary:**

This paper proposes a novel dynamic Gaussian splatting approach based on an analytical velocity field formulation. Similar to Lucas-Kanade, the method linearizes the warp function at specific time intervals and approximates it with a first-order expansion. This approach effectively regularizes scene flow, resulting in improved dynamic reconstruction performance on benchmark datasets.

**Strengths:**

- The analytical formulation of the vector field is interesting.
- More tractable than fully MLP-based Deformable 3DGS or other 2D prior-based supervision approaches.
- Competitive benchmark performance.
- The paper is clearly written and easy to follow.

**Weaknesses:**

- Limited evaluation dataset. Since the method builds on Deformable GS, it would be beneficial to evaluate on the same benchmark datasets, such as DNeRF, HyperNeRF, and NeRF-DS.
- While the method claims tractability as an advantage due to the analytical formulation, reliance on a black-box MLP weakens this claim.
- The video supplementary does not show particularly impressive results; there is noticeable unnatural blurring in the GSLK_network video

**Questions:**

- What exactly is the network architecture of the warp field?
- Compared to fully MLP-based Deformable 3DGS, grid-based warp fields such as 4DGS should struggle less with incorrect motion correlations triggered by a single global MLP representation. How much improvement does the proposed method offer for 4DGS-type warp fields?

---

> ### Author Response · Authors · 2024-11-18
> **Responses to Reviewer uwP8**
>
> We thank the reviewer for the detailed comments and for recognizing the efforts to technical novelty. We understand that there needs clarification on the network choice and our criteria for selecting benchmark comparisons. We hereby provide clarifications and will adhere to the reviewer’s suggestions in the revision.
>
> **The evaluation dataset is limited. Testing on benchmarks like DNeRF, HyperNeRF, and NeRF-DS would clarify performance relative to Deformable GS.**
>
> Our method is specifically designed for real-world scenarios characterized by minimal camera movement and the presence of highly dynamic objects. As such, testing on multi-view synthetic datasets found in D-NeRF or the highly specular dynamic scenes featured in NeRF-DS falls outside of the intended scope of our work. Instead, we focus our evaluation on datasets that are more closely aligned with our target use cases. As for evaluating our method on the HyperNeRF datasets, we currently show these quantitative results in Table 5.
>
> **Despite claims of analytical tractability, the method’s reliance on an MLP undermines this advantage.**
>
> For clarification, our method proposes a regularization of network estimated parameters, which is indifferent to different network choices. In fact, network Jacobian regularization is common in other areas of machine learning for improving prediction stabilities with perturbed inputs. Previous methods have shown that regularizing the network Jacobian, whether with domain-specific regularizers like our proposed method or through generic Frobenius norm, can enlarge the decision margin of the model and lead to more robust predictions. To prove that the method also generalizes well for other network choices, we implemented the method on the 4DGS framework. The experiment reports a PSNR value of 23.6 on the Playground scene, with a +2.2 increase from the vanilla 4DGS.
>
> **The supplementary video reveals visible unnatural blurring.**
>
> The included kid-running and train videos in supplementary were captured with cameras with very minimal translational motions, and we show comparisons with vanilla Deformable GS for comparison without our regularization. For videos like these with minimal translational motions, the rendering qualities are limited by mainly two sources of errors: accuracy of camera poses, and accuracy of trajectories. The blurring can be further improved with further optimized cameras, in addition to our motion regularizations. However, we focus on promoting scene motion integrity in our scope and suggest referring to other great works on camera pose optimization for reference (https://3d-aigc.github.io/GGRt/, https://oasisyang.github.io/colmap-free-3dgs/, https://huajianup.github.io/research/Photo-SLAM/).
>
> **Can you specify the network architecture of the warp field?**
>
> The network architecture of the warp field is identical to the one found in Deformable Gaussian Splatting.

---

### Official Review · Reviewer_GTCV · 2024-11-04

**Soundness:** 3
**Presentation:** 2
**Contribution:** 3
**Rating:** 8
**Confidence:** 3

**Summary:**

The paper addressed the problem of Gaussian Splatting estimation in dynamics scene and with small movements of the camera or the objects. The authors propose an analytical solution to the deformable warp field used to regularize the estimation instead of a learnable  field or offset-values as applied in the state of the art approaches.
The key contribution is to provide a detailed derivation and stable estimation, basically a a Gaussian Splatting warp field regularization by scene flow. The approach is then evaluated on various datasets and show improvements compared to the existing solutions.

**Strengths:**

The strength of the paper is to propose a direct analytical solution to the problem of the estimation of the warp field. The solution is mathematically transparent and provides superior results compared to a purely data-driven approach.
It is a fully new solution which is very relevant for dynamic gaussian estimation and related applications. It is strongly inspired by Lucas-Kanade point tracking and has the same level of generality.

**Weaknesses:**

- A reference implementation should be provided since the approach is very relevant to the community and may include many small details, which have their importance and not detailed in the paper.
- The method relies on external algorithms to provide depth and optical flow information, which are used to supervise the scene flow regularization. The appendix discusses experiments with different depth and flow estimation methods. However, this introduces an additional source of potential error and highlights the importance of using high-quality depth and flow estimation techniques.
- EMF is not explained when it was mentioned but later in the text.
- One reference is missing

**Questions:**

Three limitations are mentioned in the paper.
I would like to add:
- "Sensivity to noise". What is the behavior under noise? is that possible to further reduce the effect of noise?
- Computational complexity and related aspects are not mentioned at all. What is the advantage of the approach? (compared to pure learning approaches)
- The paper mentions a motion masking step. How is the threshold defined? Do theses parameters generalize?

---

> ### Author Response · Authors · 2024-11-18
> **Responses to Reviewer GTCV**
>
> We sincerely appreciate the provided valuable feedback to enhance our work’s quality! We have thoroughly considered the suggested experiments, and the suggestions on the writing. We will address these aspects appropriately and offer the following responses to the raised concerns, as well as open-source a template implementation upon acceptance to foster further studies on this end. The ambiguities in the writing will be addressed in the next revision.
>
> **What is the method’s robustness under noisy conditions, and can noise effects be further minimized?**
> The noises in the final render with dynamic Gaussian scenes are multifaceted problems that usually arise from multiple factors: perceptual noises like motion blur and lighting changes, insufficient camera motion parallax, and insufficient coverage of transient objects. We tested the method with a randomly initialized point cloud on the Dynamic Scenes dataset and reported a PSNR performance of 25.12. With the motion constraint in our approach, we have shown to improve the motion integrity of transient objects but more than such constraint on motion would be needed to tackle noises that arise from other factors. Further investigations on reducing the noise level can potentially leverage the motion uncertainties that can be derived from our Jacobian formulation.
>
> **What are the computational advantages over purely learned models?**
> We focus on geometric integrity over computational improvements. Given the time complexity bottleneck of computing the inverse of Jacobian, the approach can be further optimized by 1. employing a more efficient inversion method and 2. selectively sampling the Gaussians for regularization to reduce the complexity. Our method's memory and time consumption have been profiled as follows:
> | Method| Memory | Speed |
> | -------- | ------- | ------- |
> | Deformable GS | 10.6 GB | 14.56 it/s |
> | w/ analytical flow (100 Gaussians)| 11.3 GB  | 10.82 it/s |
> | w/ analytical flow (1000 Gaussians) | 12.7 GB  | 6.47 it/s |
>
> **How is the threshold for motion masking determined, and do these parameters generalize well?**
> The optical flow values are normalized to be ubiquitous to scene scales. The threshold of motion masking is then determined by taking statistics of flow guidances, and set at the 80 percentile of all motions, roughly at 0.1 for the “Playground” scene. Motion masking is mainly utilized for mining the Gaussians with larger motions, avoiding over-smoothing of the overall scene. We empirically ablated the choice of threshold and reported the performance differences in the following table, which will be also included in the supplementary material in the next revision.
> | Percentile | PSNR|
> | -------- | ------- |
> | 20%| 25.32    |
> | 40%| 25.86     |
> | 60%| 26.15    |
> | 80%| 26.34    |

---

### Official Review · Reviewer_GbSS · 2024-11-11

**Soundness:** 4
**Presentation:** 3
**Contribution:** 2
**Rating:** 5
**Confidence:** 4

**Summary:**

This is a paper about the fundamental question of how to derive the velocity of Gaussian means in Gaussian splatting from the finite forward warp transformation.
It assumes that these transformations are small and that one can live with a first-order Taylor expansion.

This involves the derivative of the Gaussian parameters after the motion with respect to the Gaussians in the canonical frame and the warp Jacobian.
Then, assuming that scaling and directional reflectivity do not change, one can derive the Jacobian with respect to the mean and the Jacobian with respect to the orientation.

**Strengths:**

+ Deriving analytical solutions is elegant and makes an approach interpretable.

+ The analytical derivation enables a sound application of the flow term in the Dynamic Gaussian case.

+ The paper is a pleasure to read.

+ A direct flow regularization makes a network overfit on viewpoints.

+ The idea of flow supervision in Gaussian splatting increases performance.

**Weaknesses:**

While it is nice for a paper to be self-contained, everything
in this paper until eq. (10) is identical to the "Flow supervision for Deformable NeRF" paper. Eq. (12) is the same as Eq. (7) in the latter. Optical flow rendering is same as in GaussianFlow. The reader is wondering, what exactly is the contribution of this paper when the main ingredients are in the Flow Supervision and the GaussianFlow paper. It really hurts to say that because any reader would enjoy such elegant analytical derivations. Please explain in your response.

**Questions:**

Q1: Given the analytical form of the derivation, it would be interesting to study classic optical flow problems for Gaussian flow: What is the aperture problem in the Gaussian case. What happens when the Gaussian covariance is rank deficient? What scene structures would cause an ambiguous flow?

Q2: What would happen when the spherical harmonic coefficients would change?

Q3: What happens with a local deformation like scale change or shear?

Q4: What exactly is the difficulty of including second-order terms?

Q5: How is the method's sensitivity to the choice of canonical frame? DGS methods have been proposed that do not require a choice of canonical frame.

---

> ### Author Response · Authors · 2024-11-18
> **Responses to Reviewer GbSS**
>
> We greatly appreciate the valuable suggestions and your time for the review. Your concerns help us to further improve the coherence of notations. We address the concerns in detail in the following.
>
> **While the reviewer appreciates the analytical elegance, they feel the contribution is not sufficiently differentiated.**
>
> While it may appear that our paper builds heavily on existing frameworks like [Flow Supervision for Deformable NeRF] and [GaussianFlow], we believe our work embodies a fundamental yet impactful insight. Indeed, simplicity might sometimes be mistaken for a lack of novelty. As the reviewer points out, our primary contribution lies in the reformulation of forward warp field deformation in Gaussian Splatting from the backward counterpart in NeRF, and while this may seem minor, it has significant implications for performance and generalization. With Gaussians representing explicit scene geometries, and their heavier dependency on accurate scene-to-camera projection, the deformation learned from networks can be piece-wise and even incorrectly deform supposedly static regions to compensate for photometric integrity, as shown in Figure 3 of the paper. Our approach, although simple, effectively enforces smoothness on the forward warp field, disambiguates the motions, and pushes the extent of perceptual improvements from precise motions.
>
> **How does Gaussian Flow address classic optical flow issues, such as:**
> - **The aperture problem in the Gaussian context?**
>
> Classic optical flow issues, such as the aperture problem, would likely be inherited in the Gaussian context as part of the supervision. While no existing works have directly investigated the aperture problem in the context of 3DGS, many works have discussed in the flow estimation line of work (https://openaccess.thecvf.com/content/CVPR2022/papers/Schrodi_Towards_Understanding_Adversarial_Robustness_of_Optical_Flow_Networks_CVPR_2022_paper.pdf). The optical flow map used for comparison would inherit the ambiguities arise from repetitive patterns and cause potential instabilities in the regularizations, despite the Gaussian rasterized flow map reflecting the projected motions. To explore this further, we performed experiments on the Playground scene in which different methods for flow supervision were used. We evaluated [FlowNet] and [MemFlow], achieving PSNR scores of 26.38 and 27.04. As a reference, our original setup achieved a PSNR of 26.34. [FlowNet] produced less accurate flow maps, failing to capture scene movement effectively. Conversely, [MemFlow], the current state-of-the-art optical flow estimator, achieved the best results with sharp details and accurate flow.
>
> - **Rank deficiency in Gaussian covariance?**
>
> The covariance matrices of the Gaussians express the geometry of Gaussians. Rank deficiencies can occur with needle-like or disk-shaped Gaussians, with effective ranks lower than 2. These structures, despite their potential to lead to instabilities in the inversion, are needed to represent thin and elongated geometries for certain scenes. Here with our approach, we do not set constraints on the effective rank needed for utilizing our approach for showcasing the generality. That being said, there are many great approaches for promoting higher-rank Gaussians that can be explored if needed by such particular application (https://arxiv.org/html/2406.11672v1, https://arxiv.org/html/2403.06908v1).
>
> - **Ambiguities in scene structure?**
>
> This is a valid concern, and we are actively working on a follow-up project to address ambiguities and uncertainties in scene structure, with the goal of minimizing them. Our core assumption is that deformation networks exhibit distinct patterns in regions that are easier to reconstruct compared to those that are more challenging. To explore this, we are analyzing the Jacobian of these deformations with respect to time, enabling us to quantify uncertainty directly from these patterns. In the revised Appendix Fig. A6, we present preliminary results on a controlled synthetic scene, where the red and green balls have randomized positions in the training dataset, while the blue ball accurately corresponds to the correct time step. Brighter colors indicate higher amounts of uncertainty.
>
> **How would variations in spherical harmonic coefficients affect the results?**
>
> The inclusion of spherical harmonics did not affect the results much. For the tested Playground scene, the difference only ranged within ~+0.1 PSNR.
>
>
> **What effects arise from local deformations like scale or shear?**
>
> The local deformations like changes to scale and shear would induce subtle deformations that cannot be fully captured with trajectory regularizations. To prove this point, we included the rotation and scale terms of Gaussians, which lead to a subtle increase of ~+0.2 PSNR on the tested Playground scene.

---

> > ### Author Response · Authors · 2024-11-18
> >
> > **Why are second-order terms excluded, and what challenges do they present?**
> >
> > With the second order included, the difference was minimal in terms of performance but required a longer time for each iteration, decreasing the efficiency from 10.82 it/s to 5.36 it/s.
> >
> > **How sensitive is the method to the chosen canonical frame, and could it benefit from frame-agnostic approaches as proposed by DGS?**
> >
> > The method should not be sensitive to a chosen canonical frame. As with other deformable Gaussian splatting lines of work, the perturbations in the initial point cloud would be reduced with the initial warm-up period on the static scene. To reiterate the objective, the proposed method is a motion regularization. With that said, the frame-agonistic approaches would potentially be a great outlet for our approach.

---

> > > ### Author Response · Authors · 2024-12-01
> > >
> > > Thank you for taking the time to review our work and for providing feedback. We’ve carefully addressed all the concerns raised in your comments during the rebuttal phase. If there are no remaining concerns, we kindly request you to consider revising your score to reflect the updates and clarifications we’ve made. We greatly appreciate your effort in reviewing our submission and your consideration of this request. Thank you.

---

### Author Response · Authors · 2024-11-22

**To all Reviewers:** We value all reviewers' recognition of our method’s motivation for addressing monocular dynamic Gaussian Splatting, and its strong performance. Thank you very much for the thorough reviews and feedback. We appreciate the recommended experiments and comments on the paper presentation to help refine our paper’s presentation. After carefully studying the recommendations, we included additional details about our method and supporting experiments in each discussion thread.

To summarize our approach, we present a method for regularizing dynamic Gaussian Splatting by capturing the underlying warp field dynamics. Unlike other works, our method is agnostic to the choice of deformation network and can be trained with images captured with minimal camera motions. Beyond the standard multi-view sequences, we showcase compelling performance on monocular sequences, including the NVIDIA Dynamic Scenes dataset and DAVIS dataset.

We want to thank all reviewers again and hope our additional results and rebuttal responses can clarify any doubts. We are more than happy to provide any further explanations and look forward to discussing more!

---

> ### Author Response · Authors · 2024-11-25
>
> To reviewers GbSS, GTCV, and uwP8. We have posted responses to address the concerns you have mentioned. If there are any additional concerns, we would be happy to discuss them further. Thanks!

---

### Meta-Review · Area_Chair_Fm29 · 2024-12-21

**Metareview:**

The paper presents an analytical approach to regularize dynamic Gaussian splatting using a modified Lucas-Kanade framework to compute velocity fields of Gaussian motion. It aims to improve the tractability and accuracy of dynamic scene reconstruction by using analytical formulations for scene flow instead of data-driven approaches. The proposed method achieves competitive results on monocular sequences and datasets with minimal camera motion and complex object dynamics.

Strengths:
- Analytical formulation: a novel analytical solution on velocity fields for dynamic Gaussian splatting for better tractability and interpretability.
- Practical settings: designed for realistic scenarios with minimal camera motion, where current algorithms have difficulties.
- Strong experimental results on monocular datasets and dynamic scenes.

Weaknesses:
- Unclear technical novelty: significant overlap in mathematical formulation with existing work such as "Flow Supervision for Deformable NeRF" and "GaussianFlow", raising concerns about originality.
- Limited experimental scope: evaluation on standard benchmarks such as DNeRF and NeRF-DS would better validate the generalizability of the method.
- Computational complexity: computationally intensive operations like pseudo-inverse and time integration may limit practical applications.

The paper provides a clear analytical contribution that improves the understanding and application of Gaussian splatting in dynamic scenarios. It demonstrates competitive experimental results in real-world applications with minimal camera motion.
The theoretical originality of the paper is undermined by its significant overlap with previous work, but the rebuttal partially addresses the issue.

**Additional Comments On Reviewer Discussion:**

Reviewer GbSS raised concerns about overlap with "Flow Supervision for Deformable NeRF" and "GaussianFlow" and questioned the novelty of the paper.
The authors acknowledged the overlap, but emphasized the novel application of these techniques to dynamic Gaussian splatting, with an analytical velocity formulation specifically targeting dynamic scenes.

Reviewer uwP8 suggested adding standard benchmarks such as DNeRF and NeRF-DS, and
Reviewer D8fp suggested including baselines such as PhotoSLAM for comprehensive comparisons.
The authors justified their focus on monocular sequences with minimal camera motion and indicated plans to address this in future work.

Reviewer D8fp also pointed out that explicit camera motion is not considered in the velocity field derivation.
The rebuttal stated that camera motion is implicitly accounted for in the projected optical flow, but agreed that disentangling camera motion could improve robustness and is a direction for future research.

Reviewer GTCV expressed concern about the computational cost associated with pseudo-inverse computations and time integration.
The authors provided memory and speed profiling results and suggested optimizations such as approximating the Jacobian inverse.

Reviewer uwP8 criticized the visible blurring in the supplementary videos, which undermines the results.
The blur was attributed to inaccuracies in camera pose estimation and minimal translational motion, and future improvements were suggested.

Reviewer GTCV requested a reference implementation to allow reproducibility, and the authors committed to open-source the implementation upon acceptance of the paper.

---

### Decision · Program_Chairs · 2025-01-22

Accept (Poster)